# Relationship of Body Composition and Somatotype with Physical Activity Level and Nutrition Knowledge in Elite and Non-Elite Orienteering Athletes

**DOI:** 10.3390/nu17040714

**Published:** 2025-02-17

**Authors:** Héctor Esteve-Ibáñez, Eraci Drehmer, Vladimir Schuindt da Silva, Israel Souza, Diego Augusto Santos Silva, Filomena Vieira

**Affiliations:** 1Department of Preparation and Physical Conditioning, Faculty of Physical Activity and Sports Sciences, Catholic University of Valencia “San Vicente Mártir”, 46900 Torrent, Spain; 2Federal Institute of Paraná, União da Vitória 84603-580, Brazil; vladimirschuindt@hotmail.com; 3Federal Institute of Rio de Janeiro, Paracambi 26600-000, Brazil; israel.souza@ifrj.edu.br; 4Department of Physical Education, Federal University of Santa Catarina, Florianópolis 88040-900, Brazil; diegoaugustoss@yahoo.com.br; 5Interdisciplinary Centre for the Study of Human Performance (CIPER), Faculty of Human Kinetics, University of Lisbon, 1499-002 Cruz Quebrada-Dafundo, Portugal; fvieira@fmh.ulisboa.pt

**Keywords:** anthropometry, body composition, somatotype, endurance athletes, international competition, training frequency

## Abstract

**Objectives**: The primary aim of this single cross-sectional study was to identify the physical characteristics (anthropometric, somatotype, body composition) of orienteering athletes (OAs) and to compare them with nutrition knowledge (NK) and physical activity level (PAL). **Methods**: Data were collected from 58 subjects of seven countries, including Angola (*n* = 1), Brazil (*n* = 5), Poland (*n* = 1), Portugal (*n* = 26), South Africa (*n* = 1), Spain (*n* = 22) and Sweden (*n* = 2). The subjects included 10 elite (E) female (F) OAs [age: 25.5 ± 6.4 years, body mass: 59.5 ± 7.7 kg, stature: 168.1 ± 6.5 cm, body mass index (BMI): 21.0 ± 1.9 kg/m^2^], 13 E male (M) OAs (age: 24.3 ± 5.0 years, body mass: 65.0 ± 5.5 kg, stature: 175.1 ± 6.0 cm, BMI: 21.3 ± 2.2 kg/m^2^), 18 non-elite (NE) FOAs (age: 41.7 ± 10.3 years, body mass: 60.6 ± 8.5 kg, stature: 161.3 ± 11.7 cm, BMI: 23.4 ± 3.7 kg/m^2^), and 17 NEMOAs (age: 37.2 ± 14.6 years, body mass: 71.5 ± 14.2 kg, stature: 174.0 ± 8.8 cm, BMI: 23.6 ± 4.1 kg/m^2^). The participants were selected to ensure a diverse and representative sample of international-level orienteering athletes. Measurements were taken at two IOF world ranking events, the “Portugal “O” Meeting (POM)” and the “35° Trofeo Internacional Murcia Costa Cálida”, where only top-ranked orienteers compete. The selected participants from these seven countries were among the registered athletes in these international competitions. The OAs were measured according to the guidelines of the International Society for the Advancement of Kinanthropometry (ISAK). NK was evaluated using the updated Abridged Nutrition for Sport Knowledge Questionnaire (A—NSKQ). PAL was assessed using the short version of the self-reported International Physical Activity Questionnaire—Short Form (IPAQ—SF). **Results**: The percentage of body fat (*p* < 0.01) in MOAs was significantly lower than in FOAs. Endomorphy (*p* = 0.037) and mesomorphy (*p* = 0.025) in EOAs were significantly lower than in NEOAs, but ectomorphy (*p* = 0.038) was significantly higher. EMOAs are ectomorphic mesomorphs, while NEMOAs are balanced mesomorphs, EFOAs are central, and NEFOAs are endomorphic mesomorphs. Significant differences (*p* < 0.01) were also observed in sports nutrition knowledge (SNK) among EOAs and NEOAs, with the former group achieving a higher percentage of correct responses. In the case of total nutritional knowledge (TNK), EOAs of both sexes scored significantly higher (*p* = 0.043) than their NEOA counterparts. A significant negative correlation was also observed between percentage of body fat (%BF) and metabolic equivalent (MET) in minutes per week (min/week) (*r* = −0.39, *p* = 0.038), bone mass (BM) and MET-min/week (*r* = −0.40, *p* = 0.033), and endomorphy and SNK (*r* = −0.38, *p* = 0.045) in FOAs. Among MOAs, the most significant findings included a negative correlation between age and METmin/week (*r* = −0.49, *p* = 0.010), kilocalorie (kcal) per week (*r* = −0.46, *p* = 0.016), and SNK (*r* = −0.40, *p* = 0.029). **Conclusions**: The key findings indicate that EOAs have lower BF percentages and higher NK scores compared to NEOAs. These results on the physical characteristics of OAs and the score of PAL and classification of NK can be useful to coaches and sports scientists to improve orienteer’s performance.

## 1. Introduction

Orienteering is described as a cross-country type event, with rules according to the International Orienteering Federation (IOF) [1], which manages four orienteering disciplines: (1) foot orienteering (FootO), (2) mountain bike or MTB (MTB orienteering), (3) ski orienteering (SkiO), and (4) trail orienteering (TrailO).

The physiological demands associated with FootO are comparable to those placed on long-distance runners, that is, orienteers’ performance predominantly relies on the aerobic component (which is required to maintain speed, proper technique, and mental focus), interspersed with bouts of anaerobic activity/capacity (required when energy demands are especially great, such as when running on steep uphill terrain, as well as when sprinting towards the finish) [2,3]. Given these conditions, it is hypothesized that orienteers experience high energy expenditure and substantial fluid and electrolyte losses during activity. Consequently, an appropriate physical profile and comprehensive nutritional support are essential for performance [4]. On the other hand, recent research has delved into the physiological and anthropometric traits of orienteering athletes (OAs), revealing that medalists at the World Masters Orienteering Championships (WMOCs) exhibit body mass index (BMI) values akin to those of middle-distance runners, with female athletes displaying significantly lower BMI values compared to their male counterparts [5]. Moreover, orienteers typically possess a balanced mesomorph somatotype and exhibit high levels of energy expenditure [6]. Additionally, regarding coordination abilities, FootO athletes demonstrate superior balance relative to track and field athletes [7]. Furthermore, mental fatigue induced by a 30 min cognitive task does not significantly impair the performance or physiological responses of orienteers, although a slight increase in race time has been noted [8]. These findings underscore the unique physical and cognitive demands of orienteering, indicating that success in this sport necessitates a blend of endurance, coordination, and mental resilience [3,4].

Despite orienteering’s widespread popularity across 76 countries, including Portugal and Spain, comprehensive studies that could contribute to optimizing the performance of younger elite (E) and non-elite (NE) OAs are still limited. This scarcity extends to research involving measurable human body features, which should be standardized according to the guidelines of the International Society for the Advancement of Kinanthropometry (ISAK) [9]. Additionally, there is a lack of studies assessing the physical activity level (PAL) and nutritional knowledge (NK) of these athletes [10,11,12]. A scoping review by Hopper et al. [13] emphasized that gaps in NK among athletes can lead to suboptimal dietary practices, which may hinder performance and recovery. In this sense, NK can help athletes meet recommendations for optimal energy availability and carbohydrate intake, which are crucial for maintaining performance in endurance sports [14,15]. Moreover, assessing the PAL in athletes is essential for optimizing sports performance, preventing injuries, and ensuring an appropriate training prescription. A study evaluated the PAL and energy expenditure of young athletes in aerobic sports and reported that the high level of PA observed in athletes should be considered when prescribing training to prevent overload injuries [16]. Thus, to the best of our knowledge, no studies have investigated these variables in orienteering sports. Addressing this knowledge gap could significantly improve the design of exercise/training and diet programs, as well as facilitate the monitoring of both acute and chronic effects of interventions. These elements are crucial for achieving excellence in orienteering [17].

The aims of this study were as follows:To describe the anthropometric characteristics, body composition (BC), somatotype, NK, and PAL of elite (E) and non-elite (NE) OAs;To explore the associations between BC and somatotype according to NK and PAL of E and NE OAs.

## 2. Materials and Methods

### 2.1. Ethical Approval

This project was approved by the Ethics Committee of the Faculty of Human Kinetics, University of Lisbon (Code: 13/2022) in May 2022. Informed consent was obtained from all participants included in this study. All related procedures were conducted in accordance with the standard of ethics outlined in the Declaration of Helsinki [18].

### 2.2. Study Design and Subjects

This study was conducted as a single cross-sectional study during two competitions in Portugal and Spain. A total of 58 international-level E and NE OAs of seven countries, including Angola (*n* = 1), Brazil (*n* = 5), Poland (*n* = 1), Portugal (*n* = 26), South Africa (*n* = 1), Spain (*n* = 22), and Sweden (*n* = 2), were included in this study [30 males (M) and 28 females (F); (age 41.7 ± 10.3 years: NEFOA, 25.5 ± 6.4 years: EFOA, 37.2 ± 14.6 years: NEMOA, 24.3 ± 5.0 years: EMOA)]. The main inclusion criteria for OAs were set as follows: (1) valid license in the Portuguese Orienteering Federation (FPO) [19], Spanish Orienteering Federation (FEDO) [20], or IOF [1]; (2) being in the age group of 18 to 65 years; and (3) without metabolic disease or any disease that could affect body fat and not having taken hormone treatment or corticoids in the three months prior to the anthropometric assessment, except for contraceptives. The sample size (*n*) was calculated according to the population (N) of 2500 OAs of different nationalities, considering the sum of registered participants in official, annual, and internationally renowned events in Portugal and Spain. Thus, a confidence level of 95% was adopted for the sampling calculation, as well as the associated critical value of 1.96 (Z-score), ± 10% error margin, and a population with homogeneous features (*p* = 0.8) [21], which show an “*n*” of 61 subjects. An a priori power analysis in terms of the omnibus test (ANOVA) was performed with a significant level of 0.05, a large effect size (0.4), and a power of 0.80 for four groups (2 genders and 2 elite classes), which show an “*n*” of 18 subjects for each group. A total of 45 athletes were approached, always before the competition, in each location, and the consent response rate was 25 and 33 in Spain and Portugal, respectively. Normally, the refusals occurred due to the athletes’ alleged needs to concentrate for the competition; admittedly, in orienteering, the psychological component is significant [17]. The rankings of the orienteers, separated by gender, were obtained from the IOF/World Ranking [1], accessed on 20 February 2023. An OA was classified as “NE” if they were not listed in the aforementioned rankings.

### 2.3. Variables and Measures

Sociodemographic data (i.e., date and country of birth, sex), training data (i.e., practice, frequency, and quantity), and NK and PAL data were collected from the orienteers using an online electronic scheme built on Google Forms© by the researchers. For all orienteers, anthropometric data were obtained during a single day, between 6:30 a.m. and 10:00 p.m., at two competitions in 2023: “Portugal “O” Meeting (POM)”, organized by FPO [19], and “35° Trofeo Internacional Murcia Costa Cálida”, organized by the FEDO [20] and Orienteering Federation of the Region of Murcia (F.O.R.M.) [22]. Both competitions were IOF [1] world ranking events, valid for the international ranking, in which only the best orienteers in the world compete in the elite category, who must be included in this ranking in order to participate. We consider that the COVID-19 restrictions in 2023, the year in which the data was collected, did not affect the regular orienteering schedule or training patterns, which were normal.

### 2.4. Nutritional Knowledge Assessment

To assess the NK of the OAs, especially the concepts related to sports nutrition, we used the updated Abridged Nutrition for Sport Knowledge Questionnaire (A-NSKQ) [11]. This instrument was validated for use with athletes of different nationalities, levels of competition, and sports [10,11]. The questionnaire is composed of multiple-choice questions with three or four alternative answers and just one correct answer. The questionnaire contains 35 questions, divided into two subsections. The first section contains 11 questions about general nutrition knowledge (GNK); the second section contains 24 questions specifically about sports nutrition (SNK). NK scores were expressed as percentages of correct answers obtained by the subjects in each subsection (GNK and SNK), and total nutritional knowledge (TNK) was obtained with the sum of the subsections. The level of knowledge was classified as poor (0–49%), average (50–65%), good (66–75%), and excellent (76–100%) [10]. The A—NSKQ has been shown to exhibit high construct validity (*p* < 0.001) with good test-to-test concordance (*r* = 0.80; *p* < 0.001) among athletes. From the original study, scores > 47% represent greater than average nutrition knowledge [10,11,12]. The languages used were English [23] and Spanish [24], official versions, and no cross-cultural adaptation was performed in any case.

### 2.5. Physical Activity Level Assessment

The PAL was assessed based on the short version of the self-reported International Physical Activity Questionnaire—Short Form (IPAQ—SF), in two official versions [25], English and Spanish; therefore, there was no need for cross-cultural adaptation. This instrument, with validity and re-producibility tested in numerous countries [26], consists of eight open questions that allow for estimating the time spent per week, the last seven days of the assessment, on different PA domains (i.e., walking and physical effort from moderate to vigorous intensity) and physical inactivity (i.e., sitting). Considering that the IPAQ—SF data can also be used to estimate the score expressed as metabolic equivalent (MET), in minutes per week [25,27], the total PAL score was calculated by multiplying the METs recorded for each activity type, and the volume observed for each activity type was calculated by weighting its energy requirements: walking, 3.3 METs; moderate activity, 4.0 METs; and vigorous activity, 8.0 METs. The sum of products found for each PA type gave origin to the total PAL score (walking + moderate PA + vigorous PA = total PAL score) [25]. Values lower than 10 min of PA (per day) were not included in the calculation; they were re-coded to “zero” since scientific evidence indicates that PA sessions shorter than 10 min do not lead to health benefits [25]. Cases whose total PAL score exceeded 960 min (16 h per day) were considered outliers according to the IPAQ guidelines [28]; they were excluded from the analysis. Categorical and continuous IPAQ—SF data processing and analysis followed official guidelines [25]. The PAL were categorized in three levels: (1) “low”, the lowest, for those individuals who did not walk for at least 10 min, and those who did not moderate PA were considered low/inactive; (2) “moderate”, for any of the following criteria: three or more days of vigorous activity for at least 20 min per day, five or more days of moderate-intensity activity or walking for at least 30 min per day, or five or more days of any combination of walking, moderate-intensity, or vigorous intensity activities achieving a minimum of at least 600 MET—min/week; and (3) “high”, for any of the following two criteria: vigorous-intensity activity on at least 3 days and accumulating at least 1500 MET—min/week or seven or more days of any combination of walking, moderate-intensity, or vigorous intensity activities achieving a minimum of at least 3000 MET—min/week [25].

### 2.6. Anthropometric Measurements

The anthropometric variables of the subjects were measured according to the ISAK protocol [9] by certified levels 3 and 4 Anthropometrists [29], who adopted hygienic–sanitary care against COVID-19 [30]. Twenty-six anthropometric variables were measured for each subject, namely, four basic measurements (body mass, stature, sitting height, arm span), nine skinfold thicknesses (pectoral, according to procedures described by the American College of Sports Medicine (ACSM) [31], triceps, subscapular, biceps, suprailiac, supraspinal, abdominal, front thigh and calf), nine circumferences (neck, relaxed and contracted arm, chest, waist, hip, thigh middle, calf, ankle), and four bone breadths (biepicondylar humerus, bi-styloid, biepicondylar femur, bimalleolar). Body mass was measured using a scale to the nearest 0.1 kg (Seca, model: 7601419004; Seca Gmbh & Co. KG, Hamburg, Germany), and stature and sitting height were measured using a stadiometer to the nearest 0.1 cm (Seca, model 2131721009; Seca Gmbh & Co. KG, Hamburg, Germany). Arm span was measured using a segmometer to the nearest 0.1 cm (Cescorf, Porto Alegre, Brazil). Circumferences were taken to the nearest 0.1 cm using a measuring tape (Rosscraft Innovations, Spokane, WA, USA). Bone breadths were measured to the nearest 0.1 cm using a measuring small bone caliper (Rosscraft Innovations, Spokane, WA, USA), and skinfold thicknesses were measured to the nearest 0.5 mm using a calibrated caliper (Rosscraft Innovations, Spokane, WA, USA). Repeated measures for each parameter were collected to determine the technical error of measurement (TEM) [32]. To mark the anthropometric reference points, a segmometer was used to the nearest 0.1 cm (Cescorf, Porto Alegre, Brazil), with the aid of a dermatographic pencil. BMI was calculated as body mass in kilograms divided by the square of stature in meters (kg/m^2^) [33]. Body density (BD) was estimated by using specific equations for M [34] and F [35] athletes. BD was transformed into body fat (BF) percentage using equations specific for each sex published by the ACSM [31]. Bone mass (BM) and muscle mass (MM) were determined in kilograms (kg) through the methods of Martin [36] and Lee et al. [37], respectively. Anthropometric somatotyping was performed using the Heath and Carter method [38]. Further, individual somatotypes were plotted on a two-dimensional somatochart by calculating values of *X* (ectomorphy − endomorphy) and *Y* [2 × mesomorphy − (endomorphy + ectomorphy)] coordinates: somatotype dispersion distance (SDD) (distance between mean somatoplot and each individual somatotype, represented in *Y* distance units, that is, in terms of distances at the *Y*-axis of a somatoplot), somatotype dispersion mean (SDM) (average of all the somatotype dispersion distances), somatotype attitudinal distance (SAD) (distance between any two somatopoints), and somatotype attitudinal mean (SAM) (average of the SADs of each somatopoint from the mean somatopoint). The last two are three-dimensional counterparts of the SDM [38].

Further details regarding the processes adopted for anthropometric and PAL assessment are provided in Silva et al. [6].

### 2.7. Statistical Analysis

The data collected were inputted into Microsoft Excel 2015 (Microsoft Corporation, Redmond, WA, USA). The R programming language version 4.3.1 [39] was used for all statistical analyses, including the somatoplot performed. The normality of each variable was assessed using the Kolmogorov–Smirnov normality test, and when the data were revealed as not normally distributed, non-parametric analyses were required. The measures of central tendency, namely, means (*Ms*) ± standard deviations (*SDs*) were calculated for each variable. We conduct a two-way ANOVA (Type III) in the case of normal data or Aligned Rank Transform ANOVA [40] in other cases. Post hoc multiple comparison procedures were assessed, but only the interaction term (sex–ranking) was significant; in this case, a pairwise *t*-test was used with Bonferroni correction. Pearson correlation coefficient (*r*) tests were used to calculate the relationship between the variables demographics, anthropometrics, BC, somatotype, NK, MET, and energy expenditure for PAL domain data for each sex. The results obtained following ANOVA analysis were coupled with the standardized effect size partial eta square (*η*^2^*_p_*) and were interpreted in terms of ‘small effect size’ (0.01 ≤ *η*^2^*_p_* < 0.06), ‘medium effect size’ (0.06 ≤ *η*^2^*_p_* < 0.14), and ‘large effect size’ (*η*^2^*_p_* ≥ 0.14) [41]. For the interpretation of effect size based on Pearson’s *r* values, the following guidelines were used: values around 0.10, 0.30, and 0.50 are commonly considered to be indicative of small, medium, and large effects, respectively [41]. The chosen statistical tests were appropriate for the data distribution and aimed to identify significant differences and correlations. The alpha level for statistical significance was established at *p* ≤ 0.05.

## 3. Results

A total of 58 subjects participated in this study as follows: 23 EOAs, including 10 EFOAs (age = 25.5 ± 6.4 years, body mass = 59.5 ± 7.7 kg, stature = 168.1 ± 6.5 cm, BMI = 21.0 ± 1.9 kg/m^2^) and 13 EMOAs (age = 24.3 ± 5.0 years, body mass = 65.0 ± 5.5 kg, stature = 175.1 ± 6.0 cm, BMI = 21.3 ± 2.2 kg/m^2^), and 35 NEOA, including 18 NEFOAs (age = 41.7 ± 10.3 years, body mass = 60.6 ± 8.5 kg, stature = 161.3 ± 11.7 cm, BMI = 23.4 ± 3.7 kg/m^2^) and 17 NEMOAs (age = 37.2 ± 14.6 years, body mass = 71.5 ± 14.2 kg, stature = 174.0 ± 8.8 cm, BMI = 23.6 ± 4.1 kg/m^2^). With this sample size, a power of 0.7 was achieved. The EOAs were significantly younger (*p* < 0.01) and had a higher frequency of orienteering training (OTF) (measured in days per week) (*p* < 0.01) and orienteering training quantity (OTQ) (measured in hours per week) (*p* < 0.01) compared to their NEOA counterparts. The MOAs had a significantly higher body mass (kg) (*p* < 0.01), stature (*p* < 0.01), and arm span (*p* < 0.01) than the FOAs. The EOAs had a significantly shorter sitting height (*p* = 0.018) and lower abdominal skinfold thickness (*p* < 0.01) compared to the NEOAs. The MOAs had a significantly lower triceps (*p* < 0.01), biceps (*p* < 0.01), front thigh (*p* < 0.01), and calf (*p* < 0.01) skinfold thickness compared to the FOAs. Additionally, the circumferences of the neck (*p* < 0.01) and contracted arm (*p* < 0.01) were significantly greater in the MOAs than in the FOAs, and the chest (*p* = 0.023) and waist (*p* < 0.01) were significantly smaller in the EOAs than in the NEOAs. The chest (*p* < 0.01) and waist (*p* < 0.01) circumferences were also significantly smaller in the FOAs compared to the MOAs. The percentage of body fat (BF) (*p* < 0.01) in the MOAs was significantly lower than in the FOAs. The BM (kg) (*p* < 0.01) and MM (kg) (*p* < 0.01) in the MOAs were significantly higher than in the FOAs. The BMI was significantly lower in the EOAs than in the NEOAs (*p* = 0.012). Endomorphy (*p* = 0.037) and mesomorphy (*p* = 0.025) in the EOAs were significantly lower than in the NEOAs, but the ectomorphy (*p* = 0.038) was significantly higher (Table 1, Figure 1 and Figure 2). Endomorphy (*p* < 0.01) in the MOAs was significantly lower than in the FOAs. No significant interaction between sex and ranking was found.

All significant comparisons show a large effect size (partial eta squared: *η*^2^*_p_* > 0.14), except sitting height, abdominal skinfold thickness, chest circumference, BMI, endomorphy, mesomorphy, ectomorphy, and TNK (all in EOA × NEOA comparison), with a medium effect size (partial eta squared: 0.06 < *η*^2^*_p_* < 0.14).

Although the SDM and SAM did not show significant differences, both the SDD (FOA: 4.3; MOA: 3.1) and SAD (FOA: 1.8; MOA: 1.3) exhibited values greater than 3 and 1, respectively. This indicates considerable differences in somatotypes between E and NE for the MOAs (Figure 3) and the FOAs (Figure 4). Thus, the EMOAs are ectomorphic mesomorphs, while the NEMOAs are balanced mesomorphs, the EFOAs are central, and the NEFOAs are endomorphic mesomorphs.

Significant differences (*p* < 0.01) were also observed in the SNK among the EOAs and NEOAs, with the former group achieving a higher percentage of correct responses. In the case of TNK, the EOAs of both sexes scored significantly higher (*p* = 0.043) than their NEOA counterparts.

Table 2 shows the association of NK with BC, somatotype, MET, and energy expenditure for the orienteers. The most significant findings for the FOAs include a negative correlation between age and SNK (*r* = −0.40, *p* = 0.034) and TNK (%) (*r* = −0.41, *p* = 0.030). Furthermore, OTF and OTQ exhibit a positive correlation with MET-min/week, kcal/week, GNK, SNK, and TNK, all with a *p*-value less than 0.05. A significant negative correlation is also observed between BF and MET—min/week (*r* = −0.39, *p* = 0.038), BM and MET—min/week (*r* = −0.40, *p* = 0.033), and endomorphy and SNK (*r* = −0.38, *p* = 0.045). Lastly, a positive correlation is observed between GNK and SNK (*r* = 0.50, *p* = 0.006). Among the MOAs, the most significant findings include a negative correlation between age and MET-min/week (*r* = −0.49, *p* = 0.010), kcal/week (*r* = −0.46, *p* = 0.016), and SNK (*r* = −0.40, *p* = 0.029). Furthermore, OTF and OTQ exhibit a positive correlation with MET-min/week and kcal/week with a *p*-value less than 0.05. A significant negative correlation is also observed between BF and MET—min/week (*r* = −0.48, *p* = 0.011) and kcal/week (*r* = −0.42, *p* = 0.029), MM and MET—min/week (*r* = −0.45, *p* = 0.020), BMI and MET—min/week (*r* = −0.57, *p* = 0.002) and kcal/week (*r* = −0.50, *p* = 0.008), endomorphy and MET—min/week (*r* = −0.44, *p* = 0.021), and mesomorphy and MET-min/week (*r* = −0.47, *p* = 0.013) and kcal/week (*r* = −0.45, *p* = 0.018). Ectomorphy shows a positive correlation with MET-min/week (*r* = 0.59, *p* = 0.001) and kcal/week (*r* = 0.54, *p* = 0.004). Lastly, a positive correlation is observed between GNK and SNK (*r* = 0.55, *p* = 0.002).

## 4. Discussion

### 4.1. Nutritional Knowledge and Physical Activity Levels

This study aimed to identify the physical characteristics (anthropometric, somatotype, BC) of orienteers, with rankings separated by gender, and to compare them with NK and PAL.

In sports, nutrition and performance are inextricably linked, and the assessment of NK is an important component of providing support to athletes [42,43]. For instance, Relative Energy Deficiency in Sport (RED) can be prevented through nutritional intervention in sports nutrition and/or individual athlete-centered nutrition counseling, with evidence-based information and recommendations [44,45]. Similarly, a recent intervention was conducted in female endurance athletes [46].

To the best of our knowledge, this is the first study to investigate the NK of orienteers that presented results similar to E and NE Australian team sport athletes who play Australian football, cricket, lawn bowls, soccer, or hockey [47] and E Greek handball players [48]. Therefore, although athletes may have GK of healthy eating, we understand that they may need specific knowledge of sports nutrition. Thus, we recommend that nutritionists become part of orienteering teams, as orienteers may use non-professional sources of nutrition, and studies have identified positive effects of nutritional support on NK [49,50]. We emphasize that NK is crucial for athletes to make informed dietary choices that support their training and recovery processes [13].

In general, athletes exhibit a high PAL due to the demands of their specific sport [16,51]. This was also evident in the present study, where EOAs had higher energy expenditure than NE athletes. Continuous monitoring of PAL should be considered by training teams to ensure an accurate estimation of training load, aiming to prevent overtraining in the athlete population [16].

### 4.2. Training Frequency and Risk of Injury

The data show that EOAs are significantly younger and train more frequently and for longer durations each week than NEOAs; however, approximately 500 h of annual training can present at least one injury (currently a hotly debated topic in sports sciences) in the same period of time [52], causing potential losses in performance. Younger athletes often have the capacity to train more intensively and recover more quickly [2]. However, a recent systematic review and meta-analysis also found that early specialization in a sport does not necessarily facilitate later athletic excellence, and participation patterns can differ significantly between junior and senior elite athletes [53].

### 4.3. Body Fat Percentage and Somatotype

EMOAs tend to have a lower BF percentage than NEOAs. This is consistent with the demands of orienteering as an endurance sport, where a leaner BF can contribute to better performance [54]. Similar trends were observed in another study with Turkish EOAs [55]. Additionally, our results are similar to the findings of a study that aimed to investigate morphological parameters (body mass, stature, BMI) of 50 medalists at the WMOC 2022 [5].

The somatotype data indicate that EFOAs have a more ectomorphic (lean) and less mesomorphic (muscular) somatotype than NEOA, similar to elite triathletes [56]. This could be related to the nature of orienteering (e.g., endurance and agility), which involves navigating a variety of terrains and, therefore, may favor a lighter body type [54]. Ectomorphy can be correlated with better performance results in orienteering sports [55]; however, ectomorphs have a tendency to more intensive lipolysis; consequently, it is recommended to eat a high-carbohydrate diet and combine evenly divided portions of protein and fat [57]. Conversely, sports among high-performance athletes in water, cycling, and combat sports may favor a more mesomorphic somatotype [58]. Genetic predisposition is only one of the components that, together with other factors (e.g., training, nutritional status, psychological preparation), play a key role in the successful career of athletes [59].

### 4.4. Implications for Training and Nutritional Strategies

The characteristics of successful orienteers reflect the multifaceted demands of the sport, which combines physical endurance, navigational skills, and strategic decision-making. These demands differentiate orienteering from many other sports, and further research could provide more insights into the optimal training strategies, physiological profiles, and nutritional needs for OAs.

These results can be useful for health and sports performance professionals working with orienteers and can help inform nutritional intervention strategies. However, more research is needed to further explore these relationships and to develop effective evidence-based interventions. These findings are consistent with previous studies that have demonstrated the importance of NK in sports performance [60,61,62]. Moreover, these results underscore the need for greater nutritional education and support for athletes, especially in endurance sports, where nutrition can have a significant impact on performance [63,64].

### 4.5. Strengths and Limitations

This study boasts several strengths, including the level of the athletes and the fact that the data were collected during two important tournaments in Europe. Additionally, anthropometric measurements were meticulously gathered by ISAK-certified Anthropometrists of levels 3 and 4, ensuring a high degree of accuracy and precision in the collected data [32].

Our study has some limitations. Firstly, there was no opportunity to conduct longitudinal research. We had a relatively small, yet representative, study sample size of the target population, and we recognize that, while the outline is good (±10% margin of error, *p* = 0.8), there was a relatively large margin of error, which should be acknowledged as a limitation. However, despite being underpowered, the effect size is still the change in the parameter that would be of interest to other scientists/coaches for practical significance [65]. Moreover, this study did not consider multiple external factors that influence NK among OAs, such as socio-economic status, access to resources, and cultural influences. And we acknowledge that NK alone may not translate into dietary behavior changes. Furthermore, we did not evaluate dietary intake patterns. Another limitation of this study was the use of the IPAQ—SF to estimate the athletes’ energy expenditure. Although the instrument has been validated for use in various countries with different cultural contexts, research consistently shows that when compared to direct methods of assessing energy expenditure and physical activity, the IPAQ—SF overestimates the estimates [66]. Therefore, this should be acknowledged, and future investigations with OAs should consider instruments with greater accuracy, wearable technologies for long-term monitoring, and at the event itself, and objective measures of physical activity (e.g., accelerometry).

## 5. Conclusions

Extensive materials were collected from 58 competitive OAs of different nationalities. The key findings indicate that EOAs have lower BF percentages and higher NK scores compared to NEOAs. Coaches and sports scientists can improve orienteers’ performance with the results of the present study (e.g., monitoring body composition, measuring the time and intensity of training routines, training periodization, and learning strategies about sports nutrition throughout their careers).

However, it is important to note that the cross-sectional nature of this study limits the ability to infer causality. Future research should explore longitudinal effects, with multi-seasonal data collection in larger cohorts, and handle potential confounding factors of targeted nutritional and training on orienteering performance. Longitudinal or intervention-based studies would be particularly valuable in establishing causal relationships and providing more robust evidence for optimizing the performance of orienteering athletes.

## Figures and Tables

**Figure 1 nutrients-17-00714-f001:**
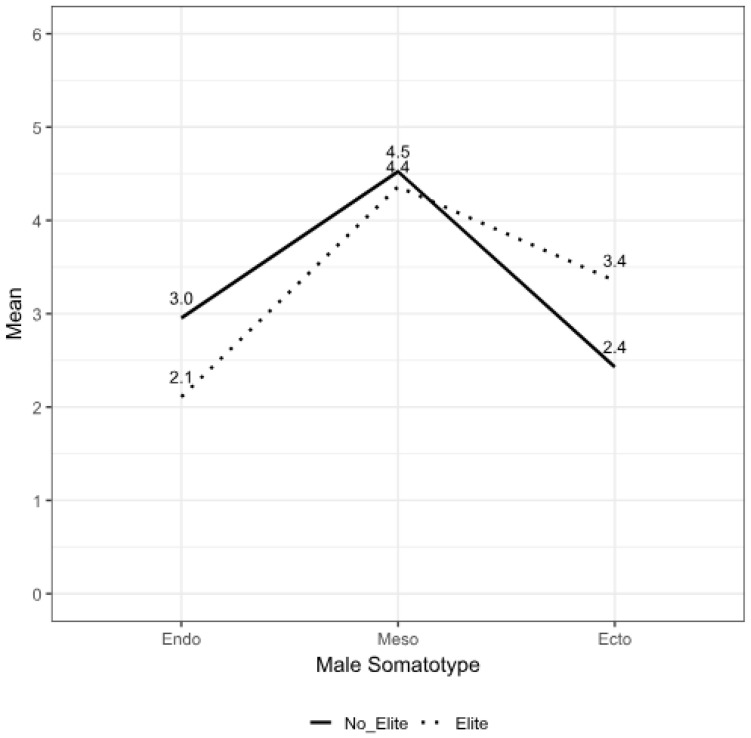
Somatotype profile distribution of non-elite and elite orienteer males.

**Figure 2 nutrients-17-00714-f002:**
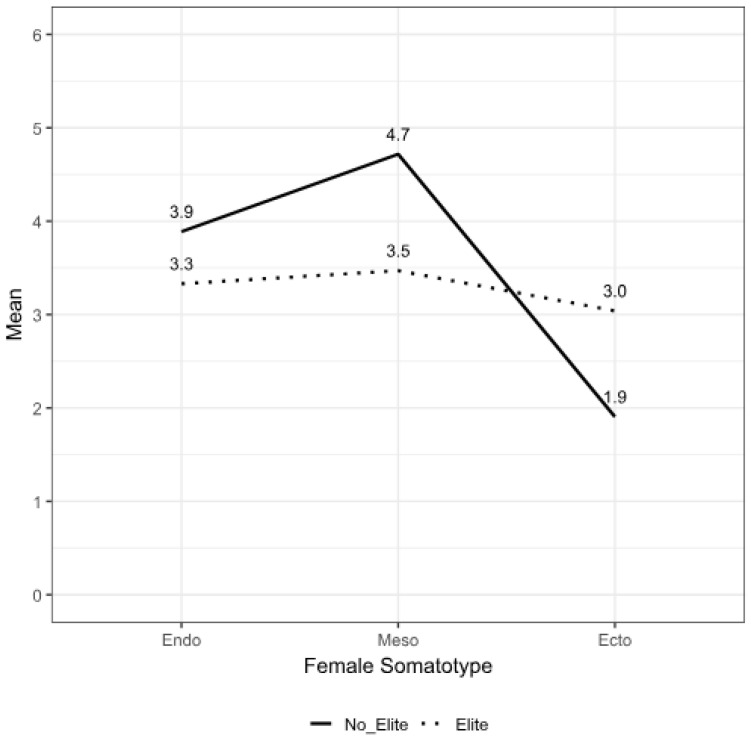
Somatotype profile distribution of non-elite and elite orienteer females.

**Figure 3 nutrients-17-00714-f003:**
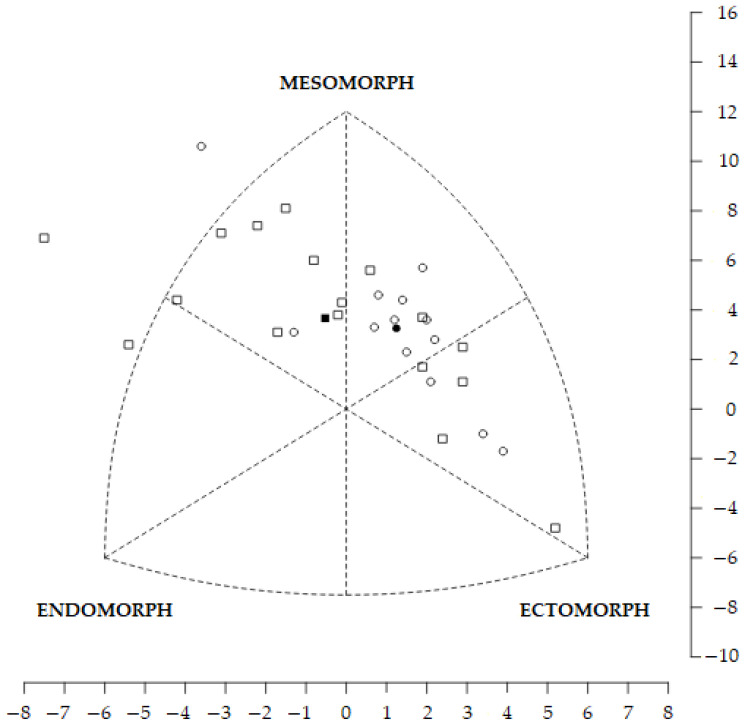
Somatotype profile distribution of male orienteers. The squares are the individual male non-elite somatotypes (the filled square is the male non-elite mean profile), and the circles are the individual male elite somatotypes (the filled circle is the male elite mean profile).

**Figure 4 nutrients-17-00714-f004:**
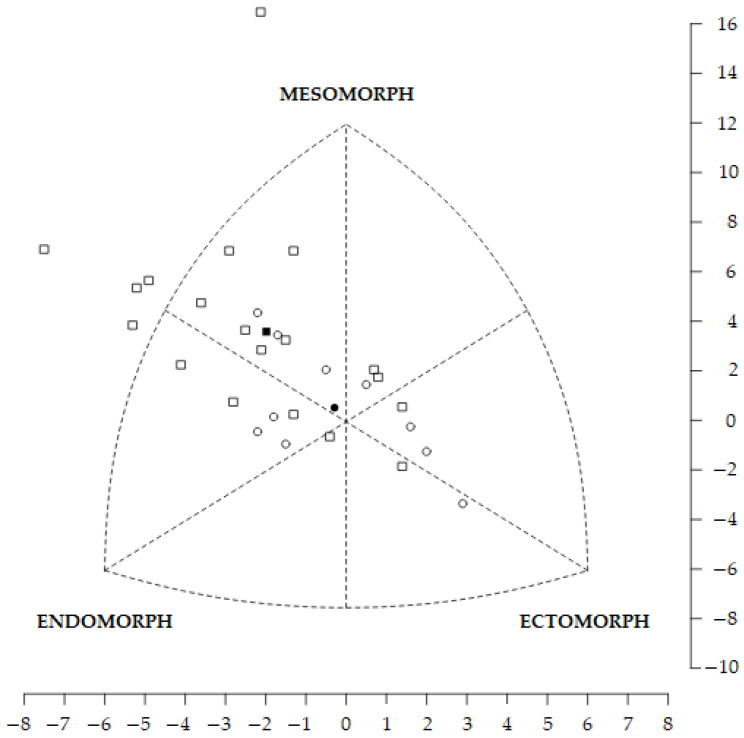
Somatotype profile distribution of female orienteers. The squares are the individual female non-elite somatotypes (the filled square is the female non-elite mean profile), and the circles are the individual female elite somatotypes (the filled circle is the female elite mean profile).

**Table 1 nutrients-17-00714-t001:** The demographic, training, anthropometric, body composition, somatotype, nutrition knowledge, metabolic equivalent, and energy expenditure characteristics, depending on ranking and sex for orienteers (*n* = 58).

Variables	Males		Females		*η*^2^*_p_* ^a^	*η*^2^*_p_* ^b^
	Elite (*n* = 13)	Non-Elite (*n* = 17)	Elite (*n* = 10)	Non-Elite (*n* = 18)		
Age (years)	24.3 ± 5.0	37.2 ± 14.6	25.5 ± 6.4	41.7 ± 10.3	0.02	0.29 *
Training						
OP (years)	12.1 ± 6.2	11.6 ± 7.6	11.3 ± 5.3	11.3 ± 9.6	0.00	0.00
OTF (days/week)	6.1 ± 1.3	3.6 ± 2.1	4.9 ± 1.6	3.0 ± 2.2	0.05	0.25 *
OTQ (hours/week)	8.8 ± 4.0	5.5 ± 5.2	7.7 ± 4.3	4.2 ± 3.3	0.04	0.18 *
Basic measurements						
Body mass (kg)	65.0 ± 5.5	71.5 ± 14.2	59.5 ± 7.7	60.6 ± 8.5	0.19 *	0.01
Stature (cm)	175.1 ± 6.0	174.0 ± 8.8	168.1 ± 6.5	161.3 ± 11.7	0.24 *	0.06
Sitting height (cm)	103.0 ± 18.7 *	122.0 ± 13.6	100.5 ± 20.2	114.0 ± 20.3	0.02	0.10 *
Arm span (cm)	176.1 ± 6.2	183.2 ± 27.0	168.2 ± 5.8	163.6 ± 7.6	0.45 *	0.02
Skinfold thickness						
Pectoral (mm)	6.4 ± 3.4	10.9 ± 9.0	8.7 ± 4.4	7.9 ± 5.7	0.00	0.00
Triceps (mm)	7.0 ± 2.8	9.2 ± 4.4	15.2 ± 5.2	15.8 ± 4.9	0.26 *	0.00
Subscapular (mm)	8.4 ± 2.6	12.1 ± 8.2	9.2 ± 2.7	11.5 ± 5.0	0.01	0.04
Biceps (mm)	3.3 ± 1.0	3.9 ± 1.6	5.3 ± 2.3	5.6 ± 2.4	0.16 *	0.02
Suprailiac (mm)	11.0 ± 7.1	15.0 ± 8.7	15.7 ± 7.3	15.4 ± 5.2	0.05	0.04
Supraspinal (mm)	7.0 ± 3.7	9.9 ± 7.8	8.1 ± 3.4	9.6 ± 4.0	0.02	0.04
Abdominal (mm)	10.1 ± 6.4	16.8 ± 9.7	13.0 ± 6.0	16.3 ± 6.2	0.02	0.14 *
Front thigh (mm)	9.2 ± 3.2	10.6 ± 5.8	23.5 ± 8.5	21.5 ± 7.3	0.53 *	0.00
Calf (mm)	5.9 ± 2.7	5.7 ± 2.5	13.4 ± 6.7	11.5 ± 4.6	0.39 *	0.00
Circumferences						
Neck (cm)	35.1 ± 1.8	37.2 ± 3.0	30.6 ± 1.6	31.6 ± 2.0	0.51 *	0.03
Relaxed arm (cm)	27.3 ± 2.4	28.8 ± 3.3	25.8 ± 1.9	27.0 ± 2.8	0.06	0.02
Contracted arm (cm)	29.1 ± 2.3	30.1 ± 3.1	26.7 ± 1.4	27.3 ± 2.1	0.18 *	0.00
Chest (cm)	90.3 ± 4.5	96.4 ± 8.5	85.5 ± 4.4	88.6 ± 6.4	0.21 *	0.09 *
Waist (cm)	74.2 ± 5.6	84.2 ± 12.1	67.2 ± 4.5	74.0 ± 7.4	0.26 *	0.21 *
Hip (cm)	91.2 ± 3.4	95.7 ± 8.2	93.4 ± 10.5	96.0 ± 7.1	0.00	0.01
Thigh middle (cm)	51.0 ± 3.1	51.1 ± 3.8	50.3 ± 3.2	49.2 ± 3.5	0.04	0.01
Calf (cm)	37.2 ± 2.1	37.4 ± 2.8	35.5 ± 2.7	36.0 ± 2.0	0.05	0.00
Ankle (cm)	22.2 ± 0.9	22.6 ± 1.8	21.8 ± 2.5	21.5 ± 1.1	0.07	0.00
Body composition						
Body fat (%)	8.4 ± 2.2	11.3 ± 4.8	17.6 ± 6.2	17.9 ± 3.9	0.27 *	0.00
Bone mass (kg)	8.7 ± 0.8	8.6 ± 1.3	7.3 ± 1.1	7.2 ± 1.0	0.21 *	0.00
Muscle Mass (kg)	30.5 ± 2.0	30.4 ± 3.4	22.1 ± 1.9	21.7 ± 2.3	0.72	0.03
BMI (kg/m^2^)	21.3 ± 2.2	23.6 ± 4.1	21.0 ± 1.9	23.4 ± 3.7	0.00	0.11 *
Somatotype						
Endomorphy	2.1 ± 0.9	2.9 ± 1.8	3.3 ± 1.1	3.9 ± 1.2	0.21 *	0.08 *
Mesomorphy	4.4 ± 1.3	4.5 ± 1.4	3.5 ± 0.9 *	4.7 ± 1.6	0.00	0.09 *
Ectomorphy	3.3 ± 1.2	2.4 ± 1.7	3.0 ± 1.0 *	1.9 ± 1.2	0.02	0.08 *
SDM	3.2 ± 3.1	5.4 ± 3.5	3.5 ± 1.6	4.6 ± 2.8	0.01	0.02
SDD	3.1		3.1		4.3	4.3
SAM	1.5 ± 1.3	2.4 ± 1.5	1.5 ± 0.6	2.0 ± 1.2	0.02	0.01
SAD	1.3		1.3		1.8	1.8
Nutrition knowledge						
GNK (%)	57.3 ± 19.4	49.2 ± 13.3	63.6 ± 19.6	54.5 ± 12.1	0.00	0.03
SNK (%)	40.4 ± 15.5	26.7 ± 15.8	47.5 ± 14.0	35.4 ± 15.7	0.06	0.15 *
TNK (%)	45.7 ± 14.3 *	33.8 ± 13.8	52.6 ± 13.4 *	41.4 ± 13.1	0.05	0.07 *
Energy expenditure						
MET—min/week	4451.3 ± 2299.4	3727.7 ± 3014.7	4249.5 ± 2346.8	3941.2 ± 2487.7	0.00	0.03
kcal/week	4948.8 ± 2520.0	4281.1 ± 2815.8	4242.3 ± 1900.7	4100.2 ± 2486.1	0.00	0.03

Data are presented as mean ± standard deviation (*M* ± *SD*); OP—orienteering practice; OTF—orienteering training frequency; OTQ—orienteering training quantity; BMI—body mass index; SDM—somatotype dispersion mean; SDD—somatotype dispersion distance; SAM—somatotype attitudinal mean; SAD—somatotype attitudinal distance; GNK—general nutrition knowledge; SNK—sports nutrition knowledge; TNK—total nutritional knowledge; MET—metabolic equivalent; Kcal—kilocalorie; min—minute. The magnitude of mean differences is expressed with standardized effect sizes. The partial eta squared (*η*^2^*_p_*) is the measure of effect size; ^a^—the *η*^2^*_p_* effect size statistic of the differences between sex characteristics of orienteering athletes; ^b^—the *η*^2^*_p_* effect size statistic of the differences between ranking characteristics of orienteering athletes; *—*p* < 0.05.

**Table 2 nutrients-17-00714-t002:** Association of nutrition knowledge with age, training, body composition, somatotype, and energy expenditure for orienteers (female, *n* = 28; male, *n* = 30).

Variables	MET—min/Week		kcal/Week		GNK (%)		SNK (%)		TNK (%)	
	F	M	F	M	F	M	F	M	F	M
Age (years)	−0.17	−0.49 *	−0.19	−0.46 *	−0.28	−0.09	−0.40 *	−0.40 *	−0.41 *	−0.34
OP (years)	−0.02	−0.28	−0.11	−0.27	−0.05	0.01	0.25	−0.07	0.17	−0.05
OTF (days/week)	0.42 *	0.40 *	0.45 *	0.37	0.41 *	0.00	0.53 **	0.33	0.55 **	0.25
OTQ (hours/week)	0.54 **	0.64 **	0.52 **	0.59 **	0.38 *	0.04	0.53 **	0.23	0.54 **	0.19
Body fat (%)	−0.39 *	−0.48 *	−0.29	−0.42 *	−0.17	−0.09	−0.25	−0.24	−0.26	−0.22
Bone mass (kg)	−0.40 *	−0.18	−0.22	−0.11	−0.13	0.15	−0.02	−0.03	−0.06	0.03
Muscle mass (kg)	−0.18	−0.45 *	−0.01	−0.37	0.08	0.25	0.13	0.13	0.13	0.18
BMI (kg/m^2^)	−0.18	−0.57 **	−0.10	−0.50 **	0.01	−0.07	−0.13	−0.22	−0.10	−0.19
Endomorphy	−0.32	−0.44 *	−0.22	−0.37	−0.21	−0.09	−0.38 *	−0.26	−0.37	−0.23
Mesomorphy	−0.12	−0.47 *	−0.08	−0.45 *	0.04	0.02	−0.10	−0.07	−0.07	−0.05
Ectomorphy	0.11	0.59 **	0.02	0.54 **	−0.01	0.12	0.27	0.21	0.21	0.20
SDM	0.06	0.19	0.01	0.21	0.10	−0.18	−0.01	−0.25	0.03	−0.25
SAM	0.07	0.13	0.02	0.15	0.12	−0.13	−0.02	−0.21	0.02	−0.21
GNK (%)	0.23	−0.09	0.25	−0.13	-	-	0.50 **	0.55 **	0.74 **	0.77 **
SNK (%)	0.15	0.15	0.10	0.10	0.50 **	0.55 **	-	-	0.95 **	0.96 **
TNK (%)	0.19	0.09	0.17	0.04	0.74 **	0.77 **	0.95 **	0.96 **	-	-
MET—min/week	-	-	0.97 **	0.99 **	0.23	−0.09	0.15	0.15	0.19	0.09
Kcal/week	0.97 **	0.99 **	-	-	0.25	−0.13	0.10	0.10	0.17	0.04

M—male; F—female; OP—orienteering practice; OTF—orienteering training frequency; OTQ—orienteering training quantity; BMI—body mass index; SDM—somatotype dispersion mean; SAM—somatotype attitudinal mean; GNK—general nutrition knowledge; SNK—sports nutrition knowledge; TNK—total nutritional knowledge; MET—metabolic equivalent; Kcal—kilocalorie; min—minute; *—*p* < 0.05; **—*p* < 0.01.

## Data Availability

All data generated or analyzed during this study are available from the corresponding author upon reasonable request. The data are not publicly available due to ethical reasons.

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
