# Peer review of "Relationship of Body Composition and Somatotype with Physical Activity Level and Nutrition Knowledge in Elite and Non-Elite Orienteering Athletes"

_nutrients, 2025, doi:10.3390/nu17040714_

Round 1

Reviewer 1 Report

Comments and Suggestions for Authors

Relationship of Body Composition and Somatotype with Physical Activity Level and Nutrition Knowledge in Elite and Non-Elite Orienteering Athletes

 Abstract:

The percentage of body fat (p = .050) in elite male was significantly lower than in non-elite male. Why does p=0.05 also indicate a significant difference between them?

 Introduction:

1.       Authors should describe more information/background about your study purpose. So the readers can understand the orienteering knowledge.

Methods:

2.       Did the authors limit the athletes’ age to involve this study? And how did the author determine the sample size?

3.       For the Abridged Nutrition for Sport Knowledge Questionnaire, how did the authors ensure that the questionnaires were consistent across countries? Are there any reliability and validity tests?

4.       For Physical Activity Assessment, since there are differences in the amount of physical activity per athlete at different levels (e.g., elite and non-elite), why measure this?

Discussion:

5.       The authors should state the limitations of this study in the discussion section.

Author Response

Dear reviewer, 

We would like to express our sincere thanks for the work you have done, thank you very much for taking the time to review this manuscript. Please find the detailed responses below and the corresponding revisions/corrections highlighted/in track changes in the re-submitted files. We have shaded in red all the changes and improvements in the article, in the revision document to your comments (attached here), our responses and arguments are in red font. We believe that we have addressed all your comments in detail. In some cases, we have endeavoured to explain the reasons for our decisions, while incorporating all your suggestions. We believe that the manuscript has been substantially improved; however, we will be happy to continue to address any additional comments you may have. 

Reviewer 2 Report

Comments and Suggestions for Authors

As a reviewer, I find the paper compelling in highlighting the role of anthropometric measures, nutrition knowledge, and physical activity in orienteering athletes. The authors have taken on a niche but increasingly popular sport, offering fresh data that could help guide training practices. However, the cross-sectional approach, small sample size, and potential variability introduced by cross-cultural differences limit the applicability of the results. Additionally, the extensive use of self-reported questionnaires—without a clear discussion of translation protocols—may introduce bias. Finally, future research would benefit from larger, more standardized cohorts, repeated measurements over time, and a clear plan for handling potential confounding factors, strengthening the conclusions drawn about Orienteering performance.

Here are detailed remarks that should be included in the new version of the paper or discussed by the authors:

(Lines 29–43): The correlation results (e.g., “r = −.39, p = .038”) are presented without clarifying the strength of these relationships in words (weak, moderate, strong). A brief qualitative interpretation may be helpful. 

(Line 10): The abstract states “58 competitive Orienteering athletes,” but does not mention the international origin (seven different countries) until the Methods. Including that detail briefly in the abstract could emphasize the study’s broader context.

Lines 45–47 7.  Some of the keywords (e.g., “body composition,” “somatotype”) closely overlap with the article’s title and main topic. Consider adding more specific terms (e.g., “endurance athletes,” “international competition,” “training frequency”) to improve discoverability.

 (Lines 48–65): While the introduction explains Orienteering as an endurance sport and references its aerobic/anaerobic interplay, it could be strengthened by citing more recent studies (post-2020) on orienteering physiology and anthropometric characteristics. 

(Lines 66–71): The introduction states that studies assessing nutritional knowledge (NK) and physical activity levels (PAL) in Orienteers are lacking. It would be helpful to include a short rationale or direct comparison to similar endurance sports (e.g., cross-country running, trail running) to justify why these findings may be unique or particularly needed. 

(Lines 72–77): The aims are stated but could be more distinctly separated (e.g., “(1) to describe… (2) to investigate correlations…”). This helps the reader see the focus of the study clearly.

(Lines 88–95): The authors mention a formula with ±10% margin of error, p=0.8, and an anticipated sample size of 61, ultimately recruiting 58. While the outline is good, 10% is a relatively large margin of error, which should be acknowledged as a limitation. Additionally, the text could clarify the response rate (e.g., how many athletes were approached vs. how many consented). 

 (Lines 81–88): The sample comes from seven different countries, but it is unclear how language or cultural differences in dietary guidelines or sports science support might impact nutritional knowledge. State whether the A-NSKQ and IPAQ-SF were used in validated translations for all participants or if only an English/Portuguese/Spanish version were used.

 (Lines 103–114): The A-NSKQ has documented validation. However, the manuscript should explicitly clarify how it was administered in multiple languages (if translations were needed) and whether any cross-cultural adaptation was performed. This matters because participants came from numerous countries.

 (Lines 116–119): The short form of IPAQ is known to overestimate or underestimate actual physical activity levels in some populations. Discussion of potential misclassification or measurement bias is warranted. 

(Lines 130–132): The manuscript excludes data >960 minutes/day. Provide the number of participants (if any) excluded for that reason, giving a transparent flow of data cleaning.

 (Lines 162–170): The study uses the equations of Withers et al. (1987) for males and females, but they are from two distinct papers (Withers 1987 for men, Withers 1987 for women). Briefly justify why these specific equations are suitable for modern orienteers, especially given they were developed with older athlete cohorts. 

(Lines 156–159): While mentioning hygienic-sanitary care is good, clarifying if COVID-19 restrictions affected the regular orienteering schedule or training patterns might be relevant to interpreting the results (especially if data were collected in 2023).

 (Lines 178–180): The authors mention Kolmogorov–Smirnov tests but do not detail the alternative if data were not normally distributed (e.g., non-parametric equivalents to the t-test, correlation). Ensure that the final analyses correctly matched the data distribution. 

(Line 183): Numerous correlations and group comparisons (elite vs. non-elite, men vs. women) are run. State if any correction (e.g., Bonferroni, Holm) for multiple comparisons was applied or justify why it was not used. 

(Line 186): Although some effect sizes are mentioned in the Results, they are not systematically reported for all significant comparisons or correlations. Including them would strengthen the interpretation of statistical significance.

 (Lines 201, 218): The text notes that BF% was “significantly” different at p = .050. This is borderline conventional significance. The authors may clarify whether p < .05 was used strictly or if p = .050 is being rounded. 

(Lines 192–193): The results mention that “EOA were significantly younger…” and trained more, but do not provide a deeper breakdown (standardizing training by total years or by weekly volume relative to body mass or other factors). 

(Lines 209–213, Figures 1–4): While the figures show the group differences, the Somatotype Dispersion (SDM, SDD) and Attitudinal distances (SAM, SAD) would benefit from a short explanation for readers less familiar with these specialized terms. 

(Lines 229–231): Many correlations are reported. The authors should highlight the strongest and the most practically significant ones (e.g., negative correlation between age and MET-min/week in men). Presenting too many correlations can dilute the main findings unless carefully interpreted.

Lines 233–323 25: The Discussion is relatively brief for the number of variables analyzed. The text could be improved by grouping results thematically (e.g., anthropometry vs. nutrition knowledge vs. physical activity). 

(Lines 233–235): Emphasize limitations of a total sample of only 58 with four subgroups (elite vs. non-elite × men vs. women). This can reduce statistical power and generalizability. 

(Lines 317–322): The discussion could be strengthened by explaining mechanisms by which higher nutritional knowledge might lead to improved performance and by acknowledging whether the cross-sectional design prevents inferring causality.

(Lines 327–330): The conclusion references the cross-sectional nature but does not sufficiently stress the lack of causal inference. A final recommendation for future longitudinal or intervention-based studies would enhance this section. 

(Lines 325–327): Briefly raised but not actually measured. If advocating that “diet quality scores or indices may be a useful tool,” the authors could recommend the specific methods or validated diet quality tools to use in future research.

 (Lines 384, 388–389, etc.): Some references mention “accessed on 12 January 2025,” which appears to be a future submission date. Or not?

Author Response

(The authors gave the same response as above.)

Round 2

Reviewer 1 Report

Comments and Suggestions for Authors

Authors have fully addressed my concerns. There are no more questions.

Author Response

There have been no revisions on their part. We welcome the comments and support from the reviewer in the first round, which undoubtedly improved our manuscript.

Reviewer 2 Report

Comments and Suggestions for Authors

Critical Scientific Review

Overall Assessment

The manuscript presents an interesting study on the relationship between body composition, somatotype, physical activity level (PAL), and nutrition knowledge (NK) in elite and non-elite orienteering athletes. While the study provides valuable insights, several areas require improvement in terms of clarity, methodology, statistical analysis, and discussion of findings. Below are detailed comments addressing specific concerns.

Major Concerns

1. Abstract and Objectives (Lines 17-49)

  • Line 19-21: The description of data collection lacks clarity regarding sampling strategy. What was the rationale for selecting participants from seven specific countries?

2. Introduction (Lines 53-98)

  • Line 57-60: The claim about aerobic and anaerobic demands should be substantiated with additional references beyond a single source.
  • Line 67-72: The description of body composition in World Masters Orienteering Championships (WMOC) competitors is interesting, but how these findings relate to younger elite or non-elite athletes is unclear. The introduction should clarify whether previous research has examined these groups separately.
  • Line 75-81: The justification for the study is weak. Compared to other endurance sports, why are PAL and NK particularly important in orienteering? Providing a stronger theoretical framework would improve the rationale.

3. Methods (Lines 99-187)

  • Line 107-130: The inclusion criteria should specify whether any specific training history (e.g., minimum years of experience) was required.
  • Line 118-122: The sample size calculation is not justified in terms of statistical power. What assumptions were made regarding expected effect sizes? It is mentioned that the study was underpowered (power = 0.7, lower than the required 0.8), raising concerns about the reliability of the findings.
  • Line 141-152: The description of the A-NSKQ is vague. How were the scoring thresholds (poor, average, good, excellent) determined? Were these cutoffs validated in prior literature?
  • Line 159-186: The International Physical Activity Questionnaire (IPAQ-SF) has known limitations, particularly in self-report bias. Were any objective measures of physical activity used (e.g., accelerometry)?
  • Line 188-224: The anthropometric assessments are thorough, but the study does not mention inter-rater reliability. How was measurement consistency ensured?

4. Results (Lines 248-347)

  • Line 249-254: The demographic breakdown is reported without statistical comparisons. Were there significant differences in age or experience between elite and non-elite groups?
  • Line 259-270: The comparisons between elite and non-elite athletes lack proper adjustments for multiple comparisons. The manuscript should indicate whether a correction (e.g., Bonferroni) was applied.
  • Line 291-294: While somatotype differences are noted, their practical significance is not discussed. Does a shift toward ectomorphy correlate with better performance outcomes in orienteering?
  • Line 322-333: The correlations between NK, PAL, and somatotype are reported without effect size interpretation. Are these relationships weak, moderate, or strong?
  • Line 335-347: Some correlations are significant, but their directionality is unclear. For example, is better NK driving better PAL, or vice versa? The text should clarify the causality implications.

5. Discussion (Lines 349-413)

  • Line 355-369: The discussion on NK is informative but fails to acknowledge that NK alone may not translate into dietary behavior changes. Were dietary intake patterns assessed?
  • Line 372-379: The injury incidence discussion is only loosely related to the study results. Since injuries were not directly assessed, this section appears speculative.
  • Line 387-398: The claim that lower body fat percentages enhance orienteering performance is made without direct performance data. Were race results or competitive rankings analyzed?

6. Limitations and Future Directions (Lines 415-432)

  • Line 421-423: The sample size limitation is acknowledged, but the authors do not propose how future research could overcome this (e.g., multi-seasonal data collection).
  • Line 427-432: The limitations of self-reported PAL are noted, but no alternatives (e.g., GPS tracking, heart rate monitoring) are suggested for future studies.

7. Conclusion (Lines 433-443)

  • Line 437-439: The conclusion states that findings can help coaches and scientists but does not offer concrete recommendations. How should training or nutrition strategies change based on the study?

Minor Concerns

  • Line 16: The title is lengthy and could be more concise without losing meaning.
  • Line 122: The phrase “a power of 0.8 calculated for determination of initial sample size” is awkwardly worded.
  • Line 146: The term “poor (0—49%)” should clarify if these cutoffs are arbitrary or evidence-based.
  • Line 176: The description of IPAQ-SF scoring lacks clarity. A brief mention of how MET-min/week categories were derived would be useful.
  • Line 246: The choice of an alpha level of 0.05 should be justified, considering multiple hypothesis testing.

Author Response

Thank you very much for taking the time to review this manuscript. Please find the detailed responses below and the corresponding revisions/corrections highlighted/in track changes in the re-submitted files. This is the second round of review, and we have put significant effort into addressing all the comments and suggestions provided. We appreciate your constructive feedback and have made the necessary adjustments to improve the overall quality of the manuscript. We have added additional references to support our claims, clarified the inclusion criteria and sample size calculations, and acknowledged the limitations of our methods. We have also provided a more detailed discussion on the practical significance of our findings and included recommendations for future research and practical applications. We hope these revisions meet your expectations and enhance the manuscript. We are attaching the file with all the addressed comments. Please note that they appear in red font in the review document. Only the new changes in the final manuscript are highlighted in red; the shading from the first round of review has been removed for clarity. 

Round 3

Reviewer 2 Report

Comments and Suggestions for Authors

Authors included all my comments